# Special Aspects in Pediatric Surgical Inpatient Care of Refugee Children: A Comparative Cohort Study

**DOI:** 10.3390/children6050062

**Published:** 2019-04-30

**Authors:** Nina K. Friedl, Oliver J. Muensterer

**Affiliations:** Department of Pediatric Surgery, University Medical Center of the Johannes Gutenberg University Mainz, Langenbeckstrasse 1, 55131 Mainz, Germany; nina-friedl@gmx.de

**Keywords:** refugee, children, anemia, burns, trauma, foreign bodies, Methicillin-resistant *Staphylococcus aureus*, Multidrugresistant gram negative bacteria

## Abstract

Background: Recently, the number of refugees in Germany has skyrocketed, leading to a marked increase in refugee children admitted to hospitals. This study describes the special characteristics encountered in pediatric surgical inpatient refugees compared to locally residing patients. Methods: Hospital records of minor refugees admitted to our department from 2005 up to and including 2015 were retrospectively reviewed. Demographic data, diagnoses, comorbidities, body mass indexes, hemoglobin values, and lengths of stay were extracted and statistically compared to local patients. Results: A total of 63 refugee children were analyzed and compared to 24,983 locally residing children. There was no difference in median body mass index (16.2 vs. 16.3, respectively, *p* = 0.26). However, refugee children had significantly lower hemoglobin values (11.95 vs. 12.79 g/dL, *p* < 0.0001) and were more likely to be colonized with methicillin-resistant *Staphylococcus. aureus* (8% vs. 0.04%, *p* < 0.01). Refugees were much more likely to present with burn injuries (16% versus 3% of admissions, *p* < 0.001), esophageal foreign bodies (4% vs. 0.5%, *p* < 0.001), as well as trauma, except for closed head injury. Conclusion: The cohort of refugee children in this study was found to be at a particular risk for suffering from burn injuries, trauma, foreign body aspirations, and anemia. Appropriate preventive measures and screening programs should be implemented accordingly.

## 1. Introduction

In the years 2014 and 2015, the number of incoming refugees in Germany skyrocketed. Approximately one-third of the arrivals were estimated to be minors [1,2], leading to a marked increase in refugee children admitted to pediatric and pediatric surgical departments. This wave has posed a variety of challenges to the healthcare system, ranging from language barriers and cultural issues to special resource allocation and funding of care [3].

We noticed that refugee children presented to our department with diagnoses and comorbidities that were markedly different than those of the native population. In particular, we were faced with a wave of burn and scald injuries at an intensity and scale not seen since the 1990s, before widespread anticipatory guidance and awareness campaigns were implemented in German society [4].

In this study, we investigated the special characteristics of refugee children admitted to a large academic pediatric surgical department in the southwest of Germany, focusing on diagnoses, colonization with resistant organisms, laboratory values, and growth parameters.

## 2. Materials and Methods

### 2.1. Ethics

The study was submitted to the ethics board of the state of Rhineland-Palatinate and deemed exempt from formal review due to the retrospective analysis of anonymous, routine medical data according to §36 and §37 of state hospital law.

### 2.2. Study Design

All patients admitted to our department from January 2005 until December 2015 were included. None were excluded. In order to discriminate between newly arrived refugee children and the local population, we stratified patient datasets by insurance status. While all children in Germany are generally insured either through private or publicly regulated companies, medical care of newly arrived refugees is funded by the municipality where the patient is registered. Once the asylum request is granted, which generally takes more than 6 months, the payer switches to one of the insurance companies. This difference is clearly marked in the electronic medical records, allowing us to discern between the newly arrived refugee children and those with permanent resident status.

Electronic medical records were then searched for demographic data, diagnostic and procedural codes, laboratory data, and findings of microbiological screening tests. These were statistically compared for differences between the cohorts.

### 2.3. Statistics

The statistical analysis was performed with SPSS for Windows 8, Version 22.0 (IBM Deutschland GmbH, Ehningen, Germany), using a *t*-test, Mann–-Whitney test, and chi-squared analysis where appropriate. A level of *p* < 0.05 was defined as significant. Values are given as means with the standard deviation after the ± symbol, except when noted otherwise.

## 3. Results

### 3.1. Demographics

The annual numbers of refugee children admitted to our department increased at least 10-fold in the years 2014 and 2015 compared to the previous years from 2005 up to and including 2013 (Figure 1). A total of 63 admitted inpatient refugee children were analyzed and compared to 24,983 locally residing children. Many patients (69%) had no information on country of origin registered in the electronic medical records. The remainder came from Syria (11%), Albania (11%), Eritrea (8%), and Afghanistan (6%).

Refugee children were significantly younger (mean age 5.6 ± 4.7 years, median four years) than local residents (mean 7.5 ± 0.4 years, median 7 years, *p* = 0.03). Males were more prevalent in both groups (refugees 55%, locals 58%), without any statistical difference. There was also no difference in length of stay (median 4 days for both) between the cohorts.

### 3.2. Colonization

Screening for colonization with drug-resistant organisms was performed in 62% of refugees (40 out of 63 refugees). Those not screened were mainly presented before 2013, when universal screening was introduced in our institution. Screening was performed by nasal, axillary, and anal swab according to hospital policy, focusing on methicillin-resistant *Staphylococcus aureus* (MRSA) and multidrug resistant gram-negative microbes (MRGN). Overall, three refugee patients (8%) of those screened from 2013 onward were positive for MRSA, and only one patient of those screened from 2013 was positive for MRGN (2%). These ratios were markedly higher than for the locally residing population (0.04% and 0.06%, respectively, *p* < 0.01).

### 3.3. Growth Parameters

There were no differences between age-adjusted weight, height, or body mass index (BMI), except that the range and standard deviation of BMI was greater in refugees (Figure 2).

### 3.4. Laboratory Parameters

As demonstrated in Figure 3, hemoglobin levels were significantly lower in the refugee cohort compared to the local population (mean 11.95 ± 0.9 g/dL versus mean 12.79 ± 0.8 g/dL, respectively, *p* < 0.001). Anemia, according to age-adjusted nomograms [4,5] was present in 16% of refugee children versus 3% of the locally resident inpatient population. No other differences in laboratory parameters were found.

### 3.5. Admission Diagnosis

There were clear differences in the rates of certain admission diagnoses recorded in the two cohorts (Figure 4). While locally residing patients were about 3 times more likely to be admitted with a closed head injury, refugee children were 5.5 times more likely to suffer burns, and 8 times more likely to have a retained esophageal foreign body or wrist injury. In addition, other trauma, such as kidney injury, ankle injury, and elbow fracture, was documented more often than compared the local population.

Concerning the burns, the most common mechanisms of injury were scald injuries. In about half of the cases, the child pulled down a pot, water boiler, or other form of container from a table or kitchen counter. The burn victims in the refugee cohort were 6 girls and 2 boys, with a combined median age of 3 years.

## 4. Discussion

Minors are a substantial portion of the refugee population around the world. In several reports, about one-third of refugees were below 18 years of age [6,7], and of those, around one-quarter were unaccompanied [8]. In the year 2015, immigration to Germany reached a peak of grossly over 2 million [9], and many of these immigrants were refugees that later applied for asylum. This wave of immigrants, who mainly originated from war-torn countries such as Syria and Afghanistan, presented a formidable challenge to the social and medical system in Germany. Most of the arrivals were initially housed in rapidly designated refugee facilities. By German law, before asylum status is granted, emergency medical care for refugees is covered by the municipality where the individual patient is accommodated, and this sometimes places an extreme economic burden and risk upon small towns and cities.

While there is an increasingly robust body of literature on the mental health of young refugees in Germany [10,11,12,13], other aspects of pediatric healthcare for refugees are less well-defined [14]. In addition, there is even less information on the surgical issues relevant to refugee children in Germany [15]. One report from Switzerland in 2015 found a high burden of infections in refugee children [16], and this was corroborated by a German study from 2016 [17]. Both studies describe the lack of systematic data on other health aspects, as well as challenges regarding legal conditions, language, and cultural competencies, and the underreporting of injuries. This is why we conducted our study on the special aspects of inpatient pediatric surgical care of minor refugees admitted over the course of 11 years, with a special focus on those that arrived during the last refugee wave during the years 2014 and 2015.

The care of this vulnerable population presents several challenges. Colonization with multidrug resistant organisms is a problem, since it requires isolation of the patient and thereby increases treatment complexity. In our study, refugee children were much more likely to be colonized than the general population. This is in line with other studies which have found similar results. In one study of 383 minor refugees at the University of Münster [18], 9.8% were colonized with MRSA, and 12.9% were colonized with MRGN. In a screening study of 119 unaccompanied minor refugees in Frankfurt [19], Enterobacteriaceae with extended spectrum beta-lactamases (ESBL) were found in 35% of cases. Higher numbers of MRGN (35%) colonization were found in a study from Mannheim [20], while MRSA colonization (7%) was similar to the rates in our population. The reasons for the higher risk of colonization with multidrug-resistant organisms are unclear, but most likely associated with the crowded living conditions in the shelter and suboptimal hygienic conditions. At this time, there is no study that has looked at overall exposure to antibiotics in refugee versus locally residing children.

Anemia is common in refugee children around the world [21,22,23,24]. This finding is most likely due to poor nutrition and iron deficiency, but other factors, such as vitamin deficiency [25], unrecognized metabolic conditions [26], infections, or hematologic disorders [27], may also play a role. Finally, the lower hemoglobin values may also reflect acute blood loss from trauma or chronic inflammation.

There are several possible causes that may explain the increased risk of refugee children suffering a burn, particularly scald injuries. Others have found a similar preponderance of burn injuries among refugee children [28]. Similarly, a Turkish pediatric hospital registered a high number of Syrian refugee children being treated from January 2016 to August 2017, with burn injury being one of the most common reasons for admission, after upper respiratory infection and gastroenteritis [29]. On the one hand, these children live in very tight conditions, in which several families use one kitchen to prepare tea or soup. In our study cohort, in one-half of the refugee children that suffered burn injuries, the mechanism was spilled hot tea or hot water for tea preparation. Additionally, in one-third of cases, the child or someone else had tripped over an electric cord that resulted in a spill of hot liquid over the child. An extensive systematic review on the epidemiology of burn injuries by Dissanaike and Rahimi corroborates our findings, in that there were strong sociocultural differences in the distribution of pediatric burn injuries, that scald injuries were the most common, and that over 50% of those injuries were associated with food preparation [30]. Hence, the crowded living space in a refugee shelter may predispose children to be in the kitchen while food is being cooked or water is being boiled. Many of the families come from cultures where hot tea is a preferred beverage. Additionally, some donated cooking equipment, including water heaters, toasters, samovars, electric grill ovens, and other items, may be old, out-of-date, or dysfunctional, and therefore not meet current safety standards. Finally, the refugee families may not have had the same exposure to child safety campaigns as the German population, in which the incidence and severity of burns has declined continuously over the last 25 years [4]. As a result, we would propose actively implementing regular burn prevention safety campaigns in all refugee shelters where children and families are accommodated.

Similar mechanisms may play a role in the relatively high incidence of esophageal foreign bodies found in our study. In the crammed environment of the refugee facilities, small objects may be left on the floor where infants and toddlers play. Most of these items are coins. However, the retention of button batteries in the esophagus may cause life-threatening complications. Thus, prevention campaigns should be implemented in refugee facilities on the dangers of leaving small objects, coins, and particularly button batteries accessible to small children.

Trauma, in general, was more common among refugee children than in the locally residing population, except for minor (in the sense of light, as opposed to severe) closed head injury. In order to understand our numbers, it must be said that in our environment, children with concussions and emesis after closed head injury are primarily admitted and observed in an effort to avoid a computed tomography of the head and the associated radiation exposure. Minor closed head trauma may not be perceived as a compelling reason to come to the hospital for evaluation by refugee families to the same extent as it is for resident parents, explaining the discrepancy observed.

Our study has several weaknesses. First, the overall number of refugee patients is limited, although the admission rates increased markedly over the last 2 years of the study interval. The mechanism of identification using payer information may have missed some children. Additionally, as there were local health clinics set up in the refugee camps by charity and volunteer organizations, only a select group of patients, most likely the more severely injured or ill, were most likely triaged to the hospital for inpatient evaluation and treatment. Refugees in general tend to use emergency care services less frequently than the general population [31]. In line with this finding, refugee children in Heidelberg, Germany, were admitted to the hospital 1.8 times more when they presented to an acute care clinic compared to the general population [32], indicating that there is a selection bias towards more serious pathology at presentation in minor refugees. Finally, the retrospective analysis of electronic patient charts may lack certain information that may be incomplete due to language and communication barriers.

Nevertheless, we believe that we evaluated a representative cohort of minor refugees that recently arrived in Germany because other pediatric surgical departments experienced similar findings, particularly regarding the high incidence of burn injuries.

In the German population, there is a relatively high level of consensus that minor refugees should have the same chances of access to education, social, and medical services as the general population [1]. In an effort to optimize the healthcare system of minor refugees, it is important to remember that the need arises spontaneously, depending on worldwide geopolitical events such as famine, drought, or war. Therefore, a general baseline preparedness must be combined with a commitment to spontaneous action [33]. It is our hope that the information presented in this study will contribute to the allocation of appropriate resources towards this vulnerable group.

## 5. Conclusions

Minor refugees in Germany have a higher risk of being anemic and being colonized by drug-resistant organisms than their peers in the local resident population. They also are more likely to suffer burn injuries and other types of significant trauma, other than minor closed head trauma. Care of these patients should focus on adequate iron intake, personal hygiene, antibiotic stewardship in the treatment of minor or viral infection to limit the emergence of drug-resistant nosocomial microbial flora, and safety measures to prevent trauma and burns. Furthermore, widespread MRSA eradication should be offered to families whose children screened positive for this. Education on how to prevent scalds and burns is particularly important in this context. We would strongly suggest implementing regular burn injury prevention and safety programs for all pediatric refugees and their families.

## Figures and Tables

**Figure 1 children-06-00062-f001:**
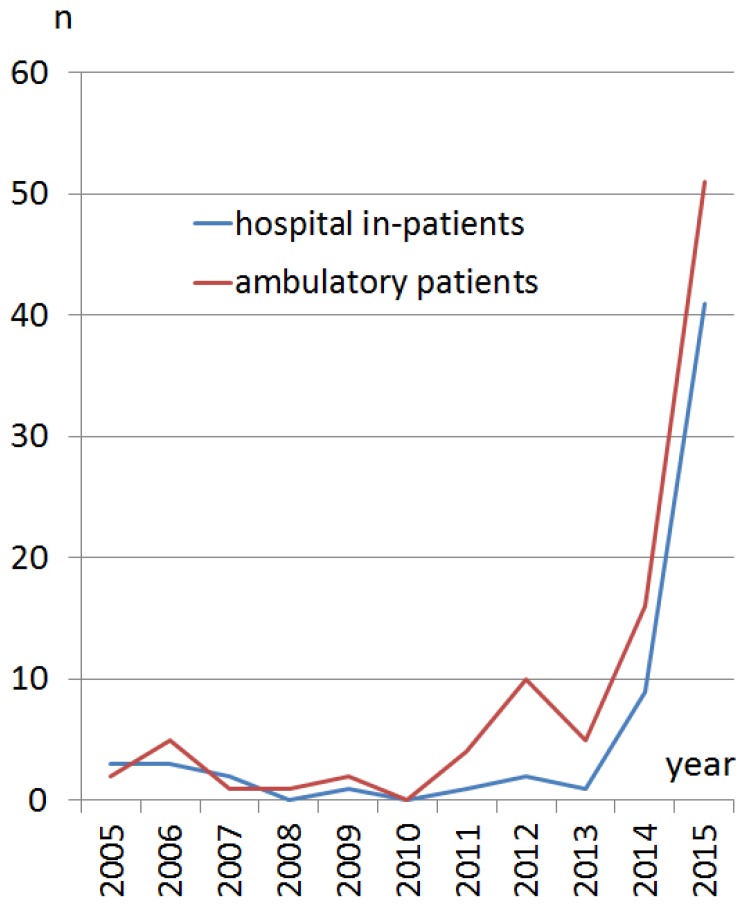
Number of refugee children treated at our department from January 2005 until December 2015. The blue line represents inpatients, the red line represents ambulatory patients. A sharp rise is evident in the years 2014 and 2015.

**Figure 2 children-06-00062-f002:**
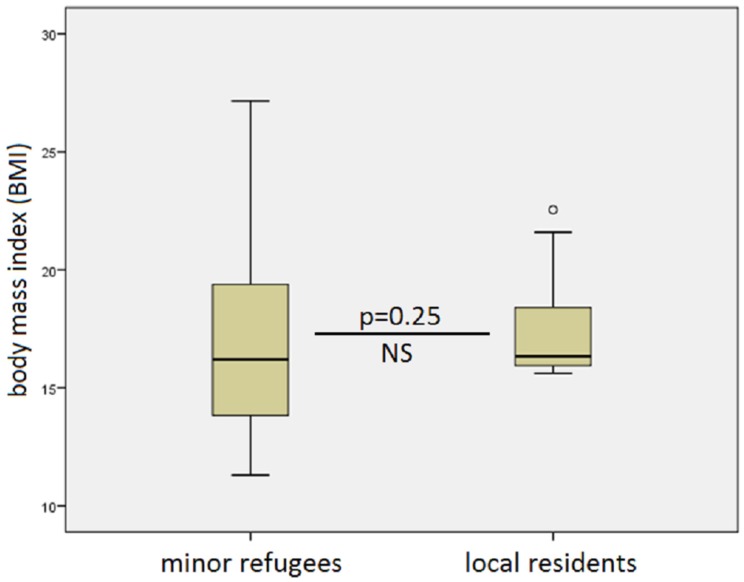
Comparison of body mass index (BMI) between refugees (left) and locally residing in-house patients. While the mean values were similar at 16.2 versus 16.3 (*p* = 0.25, nonsignificant (NS)), the range was greater in the minor refugee group.

**Figure 3 children-06-00062-f003:**
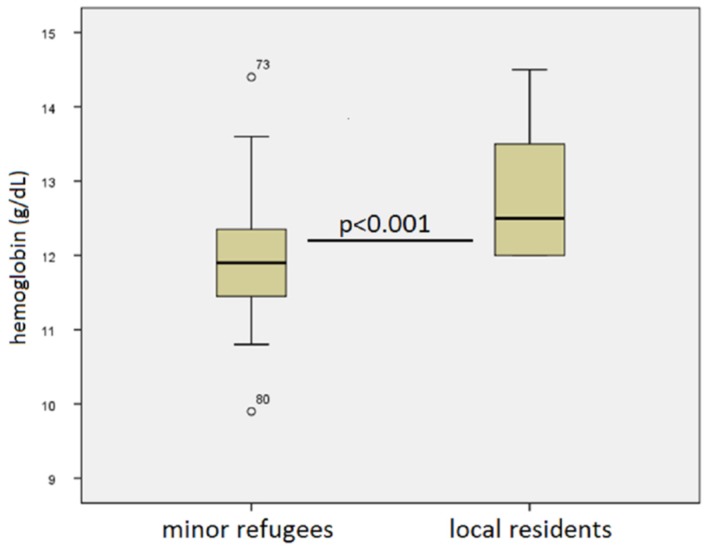
Hemoglobin values of refugee children (left) versus locally residing pediatric surgical inpatients (right). Refugees had significantly lower values and were more likely at risk for anemia.

**Figure 4 children-06-00062-f004:**
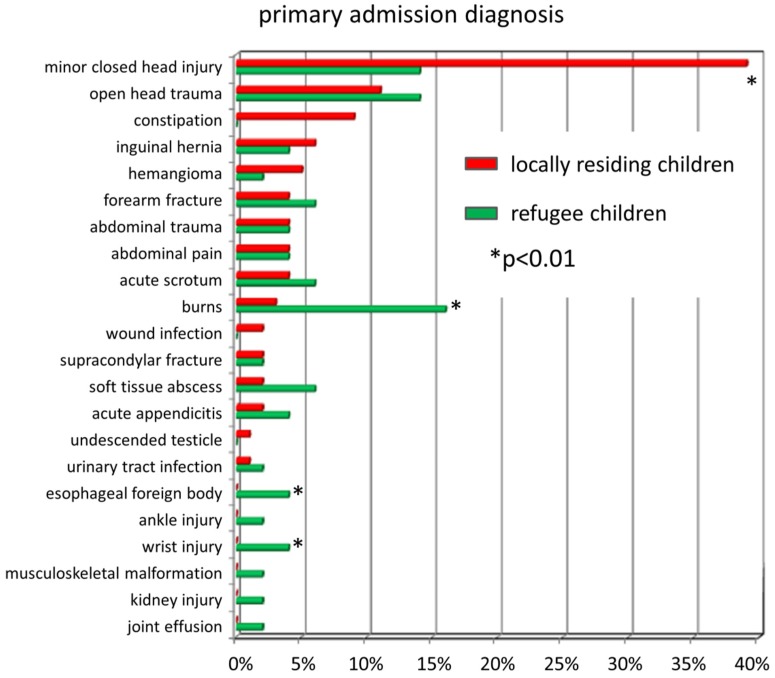
Relative proportion of admission diagnoses of refugee children (in green) and locally residing patients (in red) admitted to pediatric surgery from January 2005 until December 2015. While minor closed head injuries were more common in locals, refugees were more likely to be diagnosed with burns, esophageal foreign bodies, wrist injuries, and other types of trauma combined.

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
