# Peer review of "Special Aspects in Pediatric Surgical Inpatient Care of Refugee Children: A Comparative Cohort Study"

_children, 2019, doi:10.3390/children6050062_

Reviewer 1 Report

Line 20- I'm not so convinced you can conclude to a generalized statement that refugee children are at greater risk. Maybe for this population at this time, but I think need some caveats to even think about extrapolating to greater population.

line 28- Where is this data?

line45- You speak about the increase in the refugee pop from 2014-2015 but analyze data from 20005-2015. I would stick to one time frame. 

Line 72- Don't need this chart

Line 78- I find this confusing. I would state colonization  for the time that screening was universally performed. You can;t measure routine screening during a time that it wasn't the standard.

Line 81- You should do some more sophisticated analysis here. This does not take into consideration the n=x number of the patient populations.

paragraph that starts with line 132- I feel that you just report other results here from other papers as opposed to having a discussion

Paragraph 145 is offensive and should be removed. Refugees are not aware of dangers of scald burns to their children??!?!? I find that this paragraph demonstrates that the authors did not have any substantial contact with any patients in this cohort.

line 169- I find how they identified refugee children quite problematic 

176- typo or weird wording with the word "minor". I would keep to pediatric, and use the same "pediatric" language throughout .

190- This is interesting about the MDR organisms. Discuss this more. THIS should be a focus for the discussion section. Not a mildly offensive hypothesis about refugees not understanding fire safety. 

Author Response

1.1 Line 20- I'm not so convinced you can conclude to a generalized statement that refugee children are at greater risk. Maybe for this population at this time, but I think need some caveats to even think about extrapolating to greater population.

> The study population was specified, i.e.

"The cohort of refugee children in this study was found to be are at a particular risk for suffering from burn injuries, trauma, foreign body aspirations and anemia."

1.2 Line 28- Where is this data?

> Reference to this data was added as reference 2, the references were renumbered accordingly.

1.3 Line45- You speak about the increase in the refugee pop from 2014-2015 but analyze data from 20005-2015. I would stick to one time frame.

> We do feel that it is clear that the main increase was in the years 2014-2015, analzing the last 10 years helps put this increas into perspective.

1.4 Line 72- Don't need this chart

> We do feel that this chart shows the dramatic increase in refugees coming to our hospital, illustrating the challenges that resulted from the wave of refugees in those years.

1.5 Line 78- I find this confusing. I would state colonization for the time that screening was universally performed. You can;t measure routine screening during a time that it wasn't the standard. Line 81- You should do some more sophisticated analysis here. This does not take into consideration the n=x number of the patient populations.

> The colonization rate for the specific organisms was calculated among those screened from 2013 onwards, when universal screening was introduced. This was clarified in the sentence

"Overall, three refugee patients (8%) of those screened from 2013 onward were positive for MRSA, and only one patient of those screened from 2013 was positive for MRGN (2%). These ratios were markedly higher than for the locally-residing population (0.04% and 0.06%, respectively, p<0.01)."< span="">

1.6 In the paragraph that starts with line 132- I feel that you just report other results here from other papers as opposed to having a discussion

> We have added a discussion on the lack of robust data regarding the surgical care of refugee children and added 2 references to underline why we conducted this research.

1.7 Paragraph 145 is offensive and should be removed. Refugees are not aware of dangers of scald burns to their children??!?!? I find that this paragraph demonstrates that the authors did not have any substantial contact with any patients in this cohort. 

> The authors are pediatric surgeons with a special expertise in burn care. Every single burned individual included in this report was taken care of them personally. Our discussion does not imply that Refugees are not aware of the dangers of burns to their children. The fact that they brought their children to our hospital implies that they are very aware of the dangers and the consequences. On the contrary, we believe that the conditions in the refugee shelters predispose to a risk of burn injury, and our motivation to publish this article is to raise awareness so that these risks can be addressed. We have expanded the discussion to that regard (see Comment 2.3 to Reviewer 2).

1.8 176- typo or weird wording with the word "minor". I would keep to pediatric, and use the same "pediatric" language throughout . 

> "Minor closed head injury" is a technical term in pediatric trauma. To make it clear that we do not mean "minor in the sense of "pediatric", but "minor" as opposed to "majo"r in the sense of "light" versus "severe". This difference was described in parenthesis in the text.

1.9 190- This is interesting about the MDR organisms. Discuss this more. THIS should be a focus for the discussion section. Not a mildly offensive hypothesis about refugees not understanding fire safety.

> Unfortunately, there are no publications that analyze the reason for the higher risk of colonization with multidrugresistant organisms. We also searched for studies that looked at overall antibiotic exposure but were unable to find any. This was discussed in the paragraph.

Reviewer 2 Report

To the authors,

This is a clearly written manuscript that provides new information in the care of newly arrived refugee children.  The findings are important because they provide insight into a new area of investigation, that being the prevalence of health conditions in the hospital setting.  The methods are appropriate with the understanding of the limitations of a retrospective analysis. The conclusions are appropriate with not overstating the importance of the findings. Specifically, the increase prevalence of anemia, colonisation with Methicillin Resistant Staph aura, burns, and foreign aspiration are relevant for those who care for refugee children.

In regard to edits, I would drop the term hypothesis in the Discussion Section since not being tested to text that would speak more to the possible causes in the environment that potentially contributed to the findings as discussed in the paper.  In addition, references that would support their impressions would strengthen this section. In the Conclusion, a short paragraph that address the policy implication of the findings on the health care system, financing of care and/or resource allocation would help to link it back to this point made in the Introduction.

Author Response

2.1 This is a clearly written manuscript that provides new information in the care of newly arrived refugee children. The findings are important because they provide insight into a new area of investigation, that being the prevalence of health conditions in the hospital setting. The methods are appropriate with the understanding of the limitations of a retrospective analysis. The conclusions are appropriate with not overstating the importance of the findings. Specifically, the increase prevalence of anemia, colonisation with Methicillin Resistant Staph aura, burns, and foreign aspiration are relevant for those who care for refugee children.

> The authors thank the reviewer for their assessment.

2.2 In regard to edits, I would drop the term hypothesis in the Discussion Section since not being tested to text that would speak more to thepossible causes in the environment that potentially contributed to the findings as discussed in the paper.

> We have omitted "We have several hypotheses..." and subsituted the passage by "There are several possible causes..."

2.3 In addition, references that would support their impressions would strengthen this section.

> The entire paragraph on burn  injuries was rewritten and pertinent references were added. We also made it clearer what conclusions we draw from the high number of scald injuries in our study cohort.

2.4 In the Conclusion, a short paragraph that address the policy implication of the findings on the health care system, financing of care and/or resource allocation would help to link it back to this point made in the Introduction.

> The conclusions were expanded to include suggested concrete measures.